# Doxorubicin- and Trastuzumab-Modified Gold Nanoparticles as Potential Multimodal Agents for Targeted Therapy of HER2+ Cancers

**DOI:** 10.3390/molecules28062451

**Published:** 2023-03-07

**Authors:** Kinga Żelechowska-Matysiak, Kamil Wawrowicz, Mateusz Wierzbicki, Tadeusz Budlewski, Aleksander Bilewicz, Agnieszka Majkowska-Pilip

**Affiliations:** 1Centre of Radiochemistry and Nuclear Chemistry, Institute of Nuclear Chemistry and Technology, 03-195 Warsaw, Poland; 2Department of Nanobiotechnology, Institute of Biology, Warsaw University of Life Sciences, 02-787 Warsaw, Poland; 3Isotope Therapy Department, Central Clinical Hospital of the Ministry of Interior and Administration, 02-507 Warsaw, Poland

**Keywords:** gold nanoparticles, doxorubicin, trastuzumab, HER2+ cancers, targeted therapy, multimodality

## Abstract

Recently, targeted nanoparticles (NPs) have attracted much attention in cancer treatment due to their high potential as carriers for drug delivery. In this article, we present a novel bioconjugate (DOX–AuNPs–Tmab) consisting of gold nanoparticles (AuNPs, 30 nm) attached to chemotherapeutic agent doxorubicin (DOX) and a monoclonal antibody, trastuzumab (Tmab), which exhibited specific binding to HER2 receptors. The size and shape of synthesized AuNPs, as well as their surface modification, were analyzed by the TEM (transmission electron microscopy) and DLS (dynamic light scattering) methods. Biological studies were performed on the SKOV-3 cell line (HER2+) and showed high specificity of binding to the receptors and internalization capabilities, whereas MDA-MB-231 cells (HER2−) did not. Cytotoxicity experiments revealed a decrease in the metabolic activity of cancer cells and surface area reduction of spheroids treated with DOX–AuNPs–Tmab. The bioconjugate induced mainly cell cycle G2/M-phase arrest and late apoptosis. Our results suggest that DOX–AuNPs–Tmab has great potential for targeted therapy of HER2-positive tumors.

## 1. Introduction

Nanoparticles (NPs) are useful tools in medicine, especially in oncology, where they are applied as contrast agents for imaging techniques, radio and photosensitizers, heating agents for magnetic hyperthermia therapy, and as drug delivery systems. The mechanism of accumulation of NPs in the tumor can be either passive, the so-called enhanced permeability and retention (EPR) effect, or active, where NPs are targeted to tumor cells by vectors that possess an affinity for receptors on the cancer cells. The EPR effect exploits the preferential accumulation of NPs in the tumor through a leaky vascular system and defective lymphatic drainage in solid tumors [1]. Nanostructures, including gold nanoparticles (AuNPs) have found application in the medical field and have also become an attractive subject of scientific interest. This is due to their unique and diverse properties such as small size, universal biocompatibility, high chemical and physical stability, high drug-loading capacity, low toxicity, and versatility of functionalization. Additionally, they offer the possibility of creating innovative therapies that may provide alternatives to currently available treatments [2,3]. 

At present, multi-functionalized AuNPs are of particular interest. The delivery of an anticancer drug via NPs coupled with a specific molecule targeting the tumor can allow diagnosis with imaging functions or therapy. Traditional cancer therapies often cause significant damage in normal cells, whereas conjugation of therapeutic agents to vectors improves selectivity and enhances cytotoxicity. In particular, the use of AuNPs provides a unique opportunity to combine a number of therapeutic methods in one drug. 

Anthracyclines are one of the most widely used chemotherapeutic agents for the treatment of various types of cancers, including leukemia, Hodgkin’s lymphoma, bladder, breast, stomach, lung, ovary, and thyroid cancers, soft tissue sarcoma, multiple myeloma, and others [4] which give high responses and better survival rates [5]. They are the first anti-cancer antibiotics approved by the Food and Drug Administration (FDA), including doxorubicin (DOX), one of the first effective cytotoxic anthracycline antibiotics. Unfortunately, DOX, although an effective anti-tumor drug, displays serious side effects including rapid excretion, short retention time, and particularly cardiotoxicity as well as life-threatening cardiomyopathy and congestive heart failure [6]. To overcome these limitations [7], various types of nanostructures have been used as carriers of DOX. Liposomal-immobilized DOX (Doxil^®^) is the first FDA-approved (1995) nanodrug. Its action is based on three unrelated principles: extended drug circulation time, avoidance of RES (reticuloendothelial system) through the use of PEGylated nanoliposomes, and high and simple doxorubicin loading driven by a transmembrane ammonium sulfate gradient, which promotes drug release in the tumor. Other investigated DOX carriers include copper sulfide NPs [8], magnetite [9], and porous nanosilica [10]. However, AuNPs provide alternative approaches due to their very attractive application properties. DOX conjugation with AuNPs (DOX–AuNPs) has been previously studied and established as a water-soluble and pH-responsive anticancer drug nanocarrier. Reports have shown that DOX–AuNPs and DOX released, are localized at the perinuclear and nuclear compartments of the cells, which enhances DOX cytotoxicity. These multifunctional DOX–AuNPs are known to improve imaging contrast for photothermal cancer therapy [11]. Furthermore, DOX–AuNPs exhibit a significantly higher therapeutic anticancer efficacy (~81% tumor suppression) compared to that of free DOX (~48% tumor suppression) when tested at the same DOX dose and the same model [12]. Furthermore, tethering DOX onto the surface of AuNPs with a poly(ethylene glycol) spacer via an acid labile linkage can significantly inhibit the growth of multidrug-resistant MCF-7/ADR cancer cells, owing to the high efficiency of cellular uptake by endocytosis and subsequent acid-responsive release into cells [13].

Another advantage related to the utilization of NPs is the functionalization of their surface by numerous tumor-targeting agents such as small molecules, peptides, monoclonal antibodies, and their fragments, as well as aptamers or nucleic acids to target cancer cells, improve cytotoxicity, and further minimize side effects [1]. The most commonly used receptor-targeting agent in the treatment of patients diagnosed with HER2+ (overexpressed in approximately 25–30% of breast tumors) breast cancer is trastuzumab (Tmab). This monoclonal antibody targets HER2-positive tumors, inhibits proliferation, and finally induces cell death by extracellular and intracellular mechanisms. Currently, chemotherapy in combination with immunotherapy using antibodies targeting HER2+ receptors (e.g., Tmab and pertuzumab) is the standard care treatment improving patient survival and increasing the quality of life [14,15].

Nevertheless, many patients (50%) possess de novo resistance or acquire it after antibody treatments [16]. Therefore, the development of novel therapeutics, particularly multimodal strategies for HER2+ breast cancer to provide further clinical benefit, is highly desirable. Zhang et al. proposed the co-delivery of 5-fluorodeoxyuridine and DOX via AuNPs equipped with affibody–DNA hybrid strands to achieve a synergistic effect for the treatment of HER2 overexpressing breast cancer [17].

In this study, we investigated the synthesis and biological properties of a novel drug consisting of Tmab and DOX attached to the AuNP surface (DOX–AuNPs–Tmab), which can be potentially used in the targeted therapy of HER2+ cancers.

## 2. Results and Discussion

### 2.1. Synthesis and Characterization of DOX–PEG–AuNPs–PEG–Tmab

According to our design, AuNPs were coated with a PEG linker conjugated to DOX and Tmab. A schematic of the bioconjugate synthesis is shown in Figure 1. First, PEG (ortho-pyridyldisulfide–PEG–succinimidyl carboxymethyl ester; OPSS–PEG–NHS) comprised of a disulfide bridge with carboxyl groups at either end, was linked with a monoclonal antibody via peptide bond formation with the amine group of lysine in Tmab, which was then attached to AuNPs via a strong gold–sulfur bond [18]. In the second step, AuNPs were further PEGylated (Thiol–PEG–Carboxyl; HS–PEG–COOH) and conjugated to DOX via amide bond formation [6].

To quantify the average number of Tmab molecules conjugated to each AuNP, ^131^I-labeled Tmab was incorporated into the synthetic pathway with AuNPs. Based on radioactivity measurements, it was calculated that 72.6 ± 7.9 Tmab molecules were attached to one NP in the DOX–PEG–AuNPs–PEG–Tmab bioconjugate. The calculation took into account the mass of ^131^I-Tmab (2.55 × 10^11^ AuNPs were reacted with 4.48 ± 0.49 µg of Tmab) under the assumption that the NPs were spherical with a median diameter of 30 nm, as measured by HR-TEM, and that the density of gold was 19.28 g cm^−3^. 

Following the synthesis of AuNPs, HR-TEM measurements were performed to confirm their size and shape (Figure 2). The obtained HR-TEM images showed that the prepared NPs were almost spherical with a core equal to 30 nm. AuNP size was slightly larger (35.81 ± 0.45 nm) than that detected by TEM. DLS analysis provided information on hydrodynamic diameter and zeta potential at each step of the reaction (Table 1). The addition of PEG–Tmab and PEG–DOX to AuNPs caused an increase in hydrodynamic diameter, which confirmed the surface modification. 

The *ζ* potentials obtained for AuNPs were −45.3 ± 1.8 mV and close to −38 mV for the AuNPs–PEG–DOX and DOX–PEG–AuNPs–PEG–Tmab bioconjugates. These values revealed the high stability of the synthesized NPs and nanoconjugates and displayed no tendency to agglomerate (PDI~0.2). 

In order to determine the stability of the obtained bioconjugate in physiological solutions, namely saline (0.9% NaCl) and PBS buffer (pH = 7.4), DLS measurements were performed. Unfortunately, this method did not measure the rate of bioconjugate degradation in human serum (HS) or growing medium containing fetal bovine serum (FBS) due to the presence of large amounts of proteins, which impeded the DLS measurements. The hydrodynamic diameter of the DOX–PEG–AuNPs–PEG–Tmab bioconjugate was studied over a 14-day period (Figure 3). The results indicated that the bioconjugate remained stable in NaCl and PBS for 7 days, while mainly the diameter of PBS increased significantly 10 days from the start of the experiment. This suggested agglomeration of the bioconjugate. Nonetheless, 7 days of sufficient stability allowed efficiently targeted chemotherapy to be conducted. 

### 2.2. Binding Studies

A receptor affinity study was performed on two human cancer cell lines: HER2-positive line SKOV-3 and HER2-negative line MDA-MB-231. The binding results of the DOX–PEG–AuNPs–PEG–([^131^I]Tmab) bioconjugate are presented in Figure 4.

The obtained studies confirmed the specific binding of DOX–PEG–AuNPs–PEG–([^131^I]Tmab) to HER2+ receptors. After only 1 h incubation of the SKOV-3 cell line with the radiocompound, the percentage of binding was ~3.7%, whereas after 18 h a decrease in binding was observed. The values obtained for the negative cell line were even smaller than the measurement errors. This was due to the detected radioactivities being close to the background which confirmed the lack of specific binding to MDA-MB-231 cells. Similar results were obtained by Wawrowicz et al. [19], where receptor affinity for Au@Pt-PEG–[^131^I]Tmab was studied.

Additionally, we examined the differences in binding of the DOX–PEG–AuNPs–PEG–([^131^I]Tmab) bioconjugate to HER2 receptors on SKOV-3 cells in the presence (blocked) or absence of non-labeled Tmab (Figure 4B). The points and blue line (total) represent the sum of specific and nonspecific binding, and the pink points are values indicating specific attachment to HER2 receptors and nonspecific (green) binding to receptors that were previously blocked with a 100-fold molar excess of free Tmab. The graph for the MDA-MB-231 cell line is not shown due to the lack of specificity. These results demonstrated that DOX–PEG–AuNPs–PEG–([^131^I]Tmab) bioconjugate binds specifically to HER2 receptors expressed on SKOV-3 cells with high affinity. Our findings were in good agreement with other publications, where various NPs conjugated with Tmab were studied for receptor-binding properties [19,20,21].

### 2.3. Internalization Studies

The internalization kinetics of DOX–PEG–AuNPs–PEG–([^131^I]Tmab) in SKOV-3 cells were investigated at four time points—1 h, 6 h, 18 h, and 24 h. As shown in Figure 5, more than 97% of the bioconjugate was internalized after 1 h and remained at a comparable level for up to 24 h. Conjugation of Tmab to AuNPs did not change the internalization properties of the monoclonal antibody, which were previously confirmed [18,19,20,22,23]. Due to the lack of specific binding, no internalization study was carried out on MDA-MB-231 cells.

Furthermore, to confirm the internalization of the bioconjugate in SKOV-3 cells, a confocal microscopy imaging experiment was performed. Cells were treated with AuNPs–PEG–COOH^a^, AuNPs–PEG–Tmab, AuNPs–PEG–DOX, and finally with the bioconjugate DOX–PEG–AuNPs–PEG–Tmab. As shown in Figure 6, both AuNPs–PEG–Tmab and the bioconjugate internalized and localized in the cytoplasm of SKOV-3 cells, which was in agreement with the results obtained by the radiometric method. The less intensive dark spots observed in panels B/5 and C/5 as well as black dots in panel D/5 corresponded to AuNPs. The red fluorescence signal corresponded to DOX (panel C/4), while the green color was related to bound Tmab (panel B/3). In addition, the nuclei of their cells were stained with Hoechst 33,528 (blue fluorescence), which allowed precise observation of the localized bioconjugate. Merged signals (panels E/5, F/5) showed that DOX–AuNPs–Tmab penetrated SKOV-3 cells and localized in the perinuclear space. Moreover, the bioconjugate was detected in the nucleus, although it was rarely presented. As expected, free AuNPs (panel D/2) were internalized with low ability compared to the bioconjugate. A very low signal was detected inside the cells (panel E/2), which may suggest the passive transport of particles. The obtained results were consistent with our previously reported studies with the AuNPs–Tmab bioconjugate [19,20,21,24] as well as data published by other groups [22].

^a^ For better legibility of the legend in the following graphs, the compound designations have been changed from AuNPs–PEG–COOH to AuNPs, AuNPs–PEG–Tmab to AuNPs–Tmab, AuNPs–PEG–DOX to AuNPs–DOX and DOX–PEG–AuNPs–PEG–Tmab to DOX–AuNPs–Tmab.

Additionally, confocal imaging experiments were also performed for MDA-MB-231 cells (Figure 7), which confirmed no internalization of the bioconjugate (Figure 7B). Notably, mainly the red signal of DOX in HER2 negative cells was observed (Figure 7B), although it was presented infrequently. These results strongly indicated that the targeted bioconjugate internalizes to SKOV-3 cells (HER2 positive), whereas it did not enter cells without HER2 overexpression.

### 2.4. Cytotoxicity Studies

#### 2.4.1. MTS Assay

An MTS assay was performed on both cell lines—SKOV-3 (HER2+) and MDA-MB-231 (HER2−)—to examine the cytotoxicity. A summary of the results is shown in Figure 8. In this experiment, six various compounds including pegylated AuNPs (AuNPs), AuNPs with attached Tmab (AuNPs–Tmab), Tmab, pegylated NPs with attached DOX (AuNPs–DOX), free doxorubicin, and the bioconjugate (DOX–AuNPs–Tmab) were tested. The DOX content of NPs was adjusted to the free DOX concentration, which was 2, 7, and 15 µg/mL (Figure 8 and Appendix A).

The results showed a decrease in metabolic activity of cancer cells which depended on DOX concentration and incubation time; nonetheless, AuNPs–DOX, DOX, and DOX–AuNPs–Tmab were more toxic toward the SKOV-3 cell line than MDA-MB-231 (Figure 8). Plain NPs, Tmab, and AuNPs–Tmab were used as negative controls due to the lack of, or a slight cytotoxic effect. A comparison of the toxicity of AuNPs–DOX, DOX, and the bioconjugate revealed that the free chemotherapeutic exhibited the strongest effect. After 48 h of incubation time (Figure 8A), 69% of HER2-positive cells treated with AuNPs–DOX were detected as metabolically active, whereas for DOX it was only 29%. This effect was probably related to the blocking of AuNP (30 nm) for DOX entry into the cell nucleus, which was necessary for effective DOX action. Significant differences in SKOV-3 cells treated with AuNPs–DOX and DOX–AuNPs–Tmab after 72 h and at concentrations of 7 and 15 µg/mL were observed. The surviving fractions of the AuNPs–DOX conjugate were 57 ± 10% and 35.9 ± 1.7%, whereas for DOX–AuNPs–Tmab they were 45.0 ± 6.1% and 23.2 ± 2.5%, respectively. These results demonstrated the importance of attached Tmab in targeting DOX. Additionally, at shorter incubation times—24 h and 48 h—cell survival rates for both AuNPs–DOX and DOX–AuNPs–Tmab conjugates were comparable. Moreover, in the case of MDA-MB-231 cells, the cytotoxicity of the above-mentioned conjugates was comparable, even after 72 h, which was due to the lack of HER2 receptors on the cells. (Figure 8B). Interestingly, the obtained results were in contrast to those presented by Spadavecchia et al. [25] where the therapeutic efficacy of DOX attached to AuNPs resulted in a 30-fold decrease of EC_50_ in relation to this molecular drug. Furthermore, similar toxicity of AuNPs–PEG–DOX and DOX on A549 and HeLa cell lines was observed by Święch et al. [6]. 

Studies using folate-modified PEG-functionalized AuNPs (FOL-aPEG-GdNP) showed differences between the cytotoxic effects of FOL-aPEG-GdNP and DOX depending on the cell line tested (A549, KB, and HFF) [26]. However, the toxicity effect of DOX-loaded NPs was as high as that of free DOX. In another study [27], the effects of DOX-loaded PEI-enhanced HSA nanoparticles were compared with free doxorubicin on MCF-7 cells. The experiments showed that the cytotoxicity of doxorubicin-enhanced nanoparticles and doxorubicin alone was comparable (after 48 h, with increasing DOX concentration—0.01–100 µg/mL). Nevertheless, a decrease in cell viability at a longer incubation time—144 h and a DOX concentration of 1 µg/mL—was higher for DOX–NPs. Based on literature data and in contrast to our studies, lipid NPs loaded with DOX exhibited higher cytotoxicity and required lower concentrations of chemotherapeutic to inhibit the proliferation of cancer cells than free DOX [28]. 

By contrast, but in agreement with our results, it has been reported that chemotherapeutic agents (DOX or epirubicin) were more efficient than those synthesized with NPs [29,30]. Variations in the results may be due to different cancer cells used or some fluctuations in the DOX amount conjugated to AuNPs. 

#### 2.4.2. Apoptosis

In the next step, the cytotoxic effect of the synthesized compounds was examined by the flow cytometry method using annexin V-FITC and PI fluorescence staining assay (Figure 9, Appendix A). The apoptosis assay confirmed the non-toxic properties of AuNPs, AuNPs–Tmab, and Tmab at given concentrations. Furthermore, these results strongly indicated that cell death through necrosis or late apoptosis was mainly induced via free DOX and was both dose- and time-dependent.

In the case of the bioconjugate, after 72 h of incubation a similar percentage of late and early apoptotic cells was detected (late apoptosis—13.4 ± 1.0%, early apoptosis—14.9 ± 0.40%), whereas for free DOX, late apoptosis was dominant (37.50 ± 0.10% vs. 18.50 ± 0.30%). AuNPs–DOX was not as effective as DOX or the bioconjugate in inducing apoptosis. Similar correlations were also observed for the higher dose of DOX—15 µg/mL (Appendix A). The obtained results revealed that cell death after treatment with AuNPs–DOX, DOX, and bioconjugate occurred mainly via late apoptosis. Similar observations were presented in Nieciecka’s paper [29], where the effect of SPION@CA_DOX was tested on the SKOV-3 cell line. The authors used a lower concentration of doxorubicin to test apoptosis—0.4 µg/mL. However, the effect of free doxorubicin, in agreement with the MTS assay, was stronger than that of SPION@CA_DOX. The tested compounds also induced mainly late apoptosis.

### 2.5. Cell Cycle Assay

In order to better characterize the effects of the synthesized compounds on HER2-positive cells, cell cycle analysis was performed (Figure 10 and Appendix A). After 24 h of incubation, a significant increase in the proportion of cells was observed in the G2/M phase for free DOX or attached cytotoxic drug to AuNPs in comparison to AuNPs, AuNPs–Tmab, and monoclonal antibody, as well untreated cells. Moreover, the highest G2/M-phase arrest was detected after the treatment with the bioconjugate. In all cell lines, the G1 phase was reduced after the incubation of SKOV-3 cells with DOX compounds. A similar trend was previously reported, where the anti-Kv11.1-pAb antibody and DOX were attached to AuNPs [25]. After 48 h, an increase in the G0/G1 phase was observed, and subsequently, cells started to accumulate in the S phase up to the next measurement point (72 h). DOX, which interacts with DNA and inhibits topoisomerase II, caused cell cycle arrest in the S and G2/M phases [25,31], whereas cells treated with Tmab underwent arrest during the G1 phase [32]. Overall, taking into account G2/M and S cell cycle arrest, and the reduction in the G1 phase, our results were in good agreement with the induction of apoptosis after treatment due to DNA damage.

### 2.6. Cytotoxicity Studies on Cell Spheroids

Contrary to monolayer cultures, three-dimensional (3D) cells better mimic tissue physiology and exhibit characteristics of poorly perfused tumors. They are relevant models for examining the efficacy of anti-cancer drugs [33,34], and, therefore, our cytotoxicity studies were also carried out using spheroids (Figure 11, Figure 12, and Appendix A). The experiment was performed with a 7 µg/mL DOX concentration.

Microscopic images of the spheroids treated with the synthesized compounds and their measured areas for 30 days are presented in Figure 11 and Appendix A. Figure 12 clearly shows that unbound DOX and the bioconjugate exhibited a toxic effect. The surface area of spheroids treated with DOX–AuNPs–Tmab started to decrease after day 2, and after day 30 the area was reduced almost by half (day 0: 111 065 ± 310 µm^2^ vs. day 30: 65 763 ± 589 µm^2^). A similar, although slightly stronger effect, was achieved by DOX. Additionally, the spheroid exposed to the bioconjugate shrunk, while treatment with DOX promoted the decomposition of small cell compartments. In contrast to the bioconjugate, no changes in spheroids treated with AuNPs, Tmab, and AuNPs–DOX were observed, regardless of their initial size. Notably, the spheroids after the addition of AuNPs–DOX grew slightly (surface area on day 30 increased by only 30%), whereas untreated 3D cell culture models doubled their surface area. The obtained results were consistent with cytotoxicity studies (MTS/apoptosis assay) described in Section 2.4.

Furthermore, to confirm the effect of the tested compounds on spheroids, they were subjected to propidium iodide (PI) and Hoechst 33,258 (Hoechst) staining. Fluorescence microscopy images showed spheroids on three measurement days (days 1, 10, and 20) and are shown in Figure 12. Hoechst 33,258 dye crosses the cell membrane, and therefore it labels the DNA of both dead and living cells. On the other hand, propidium iodide penetrates cells with damaged plasma membranes and consequently stains only dead cells [35]. Based on the knowledge that DOX intercalates into DNA [36] and is a fluorogenic compound with an excitation wavelength of 490 nm and an emission of 590 nm, images were made using the settings for FITC, demonstrating a similar wavelength range.

On the first day after treatment, the strongest signals were detected for the bioconjugate and DOX indicating necrotic or late apoptotic cells. However, more intense colors were observed for DOX. In the case of other compounds studied, no signals of propidium iodide or FITC were observed at this time point. Furthermore, on day 30, the treatment with only AuNPs, Tmab, and AuNPs–DOX showed less intense red and green signals compared to DOX–AuNPs–Tmab and unbound DOX, suggesting lower cytotoxicity. 

These findings were in good agreement with MTS assay and flow cytometry experiments, where the highest cytotoxicity and cell cycle block in the G2/M and S phases for DOX and targeted AuNPs–DOX were observed.

## 3. Materials and Methods

### 3.1. Chemical Reagents

The following chemical reagents were used: gold (III) chloride trihydrate (HAuCl_4_·3H_2_O), sodium hydroxide (NaOH), trisodium citrate dihydrate (C_6_H_9_Na_3_O_9_), HS–PEG–COOH (poly(ethylene glycol), 5 kDa), doxorubicin hydrochloride form Millipore Sigma (St. Louis, MO, USA), OPSS–PEG–NHS (alpha-pyridyl-2-disulfid-omega-carboxy succinimidyl ester poly(ethylene glycol), 5 kDa) from Creative PEGworks (Chapel Hill, NC, USA), isolated Tmab from Herceptin^®^ (Roche Pharmaceuticals, Basel, Switzerland), N-(3-dimethylaminopropyl)-N-0-ethylcarbo-diimide hydrochloride (EDC, >99%) and iodogen (1,3,4,6-tetrachloro-3R,6R-diphenylglycouril) from Thermo Fischer Scientific (Waltham, MA, USA), and PD-10 column (GE Healthcare, Piscataway, NJ, USA). All aqueous solutions were prepared using ultrapure deionized water 18.2 MΩ·cm (Hydrolab, Straszyn, Poland).

### 3.2. Cell Lines and Reagents for Biological Studies

SKOV-3 (HER2-positive) and MDA-MB-231 (HER2-negative) cell lines were obtained from American Type Culture Collection (ATCC, Rockville, MD, USA) and cultured according to the ATCC protocol (humidified atmosphere of 5% CO_2_ at 37 °C). Cells were maintained in McCoy’s 5A and Dulbecco’s modified Eagle’s medium (DMEM) mediums enriched with 10% heat-inactivated fetal bovine serum and antibiotics: penicillin and streptomycin (100 IU/mL). Trypsin EDTA solution C (0.25%) was used to detach the cells. All reagents mentioned above were purchased from Biological Industries (Beth Haemek, Israel). CellTiter96^®^ Aqueous One Solution Reagent (MTS compound) from Promega (Mannheim, Germany) and phosphate-buffered saline (PBS) from Biological Industries (Beth Haemek, Israel) were used for in vitro studies. To prepare the samples for flow cytometry, reagents such as 10× Annexin V Binding Buffer, FITC Annexin V, Propidium Iodide Staining Solution (PI) from BD Biosciences (Becton, Dickinson and Company, Franklin Lakes, NJ, USA), and ethanol (ChemPur, Piekary Śląskie, Poland) were used. Hoechst 33,258 (Ph-enol, 4-[5-(4-methyl-1-piperazinyl)[2,5’-bi-1H-benzimidazol]-2’-yl]-,trihydrochloride), and Triton™ X-100 Surfact-Amps™ Detergent Solution were purchased from Thermo Fischer Scientific (Waltham, MA, USA), and Anti-Human IgG (γ-chain specific; F(ab′)2 fragment–FITC antibody was produced from goat; F1641, Millipore Sigma, St. Louis, MO, USA) were used to prepare samples for confocal microscopy. The same dyes were used for spheroids. Other reagents for confocal microscopy included paraformaldehyde (PFA) and bovine serum albumin purchased from Millipore Sigma (St. Louis, MO, USA), and DAKO Fluorescent Mounting Medium purchased from Agilent Technologies (Santa Clara, CA, USA).

### 3.3. Radionuclide

No-carrier-added ^131^I was used for the radioiodination of Tmab. Na^131^I_aq,_ in carbonate buffer, with specific activity >550 GBq/mg, was purchased from the Radioisotope Centre POLATOM (Świerk, Poland). ^131^I is a β^−^ emitter with the half-life of 8.03 days and energy 606 keV (90%), which decays to stable ^131^Xe. This radionuclide also emits high energy gamma radiation (364 keV; 10%) that can be used for imaging. 

### 3.4. Characterization Techniques for NPs

The size and shape of AuNPs were characterized by HR-TEM microscopy (TALOS™ F200X, Thermo Fischer Scientific-Waltham, MA, USA). Size and zeta potential were determined using dynamic light scattering (Zetasizer Nano ZS DLS, Malvern, UK). Thin layer chromatography was conducted using a Storage Phosphor System Cyclone Plus (Perkin Elmer, Waltham, MA, USA), glass microfiber chromatography paper impregnated with silica gel (iTLC SG, Agilent Technologies, Santa Clara, CA, USA), and PBS buffer as mobile phase, for the stability of [^131^I]Tmab. A Jasco V-650 was used for UV–vis measurements.

### 3.5. Synthesis of AuNPs

The synthesis of AuNPs was performed according to the method described in [37]. Briefly, 0.1 M NaOH (~400 µL) was added to 12.6 µmol HAuCl_4_·3H_2_O until pH ~4.5 was achieved, and then heated for 30 min at 95°C. Sodium citrate dihydrate was then added as a reducing agent (340 mM, 170 µL) and heated for a further 3 h, resulting in a color change from yellow to red [19]. The reaction mixture was cooled to room temperature, and the size, zeta potential, and shape were measured. Assuming a spherical shape of NPs with a mean diameter of 30 nm and knowing the mass of gold used in the synthesis, AuNP concentration was determined.

### 3.6. Synthesis of DOX–PEG–AuNPs–PEG–Tmab

Conditions for Tmab bioconjugation were based on the methods described in [18,19], whereas synthesis with DOX was described in [6]. A 25-fold excess of OPSS–PEG–NHS (5 kDa) in carbonate buffer (100 mM, pH 8.9) was added to Tmab (200 µg) and mixed overnight at room temperature. Excess OPSS–PEG–NHS was removed using Vivaspin^®^500 100 kDa cut-off centrifuge concentrators (Sartorius, Goettingen, Germany). An amount of 1.43 mL AuNPs in carbonate buffer (20 mM), were concentrated via the addition of 7 µg OPSS–PEG–Tmab and the reaction continued for a further 30 min in Protein LoBind Tubes (Eppendorf, Hamburg, Germany). To remove unconjugated proteins, AuNPs–PEG–Tmab was centrifuged (6500 rpm, 20 min) and dispersed again in a carbonate buffer. In the next step, 15,000-molar excess of HS–PEG–COOH (5 kDa) was conjugated for 30 min. Centrifugation was repeated and DOX hydrochloride, EDC (118 nmol dissolved in water), and NEt_3_ (0.58 µm; 80 µL from a previously prepared solution of 1 µL NEt_3_ added to 1 mL of water) were added. The reaction was carried out overnight, and then the unconjugated products were centrifuged and removed from the vial. Further, the conjugated product was resuspended in 1 mL of ultrapure deionized water. The efficiency of doxorubicin coupling was determined by UV–vis spectroscopy measurements. A calibration curve was determined, by which the concentration of the unbound chemotherapeutic agent in the supernatant was calculated. The efficiency was 95.0 ± 3.5%.

### 3.7. Stability Studies

The bioconjugate was centrifuged and dispersed in 0.01 M PBS buffer and 0.9% saline to determine its stability under physiological conditions. The solution was allowed to stand at 37 °C for 14 days and the tendency to aggregate was determined with hydrodynamic diameter measurements using the DLS method.

### 3.8. Quantification of the Number of Tmab Particles Conjugated to AuNPs

To calculate the number of Tmab molecules conjugated to each AuNP, Tmab was iodinated ([^131^I]Tmab) using the method described in [38]. Briefly, 1.5 mg Tmab and 0.01 M PBS were labeled with radionuclide ^131^I (10–20 MBq) using tubes with dried iodogen for 10 min. The product was then purified using PD-10 columns (Sephadexem G-25 resin; Millipore Sigma, St. Louis, MO, USA) with 0.01 M PBS as the mobile phase. The subsequent synthetic steps were the same as for isotope-free synthesis. To calculate the efficiency of protein conjugation with NPs, the supernatant was collected into a separate vial after each centrifugation, and the activity of each collected fraction was measured. This was determined by counting the number of iodinated Tmab per AuNP (η = 63.9 ± 6.8%).

### 3.9. Binding Studies

One day before the experiment, SKOV-3 cells (6 × 10^5^) were seeded in 6-well plates (TPP, Switzerland) and stored in a cell incubator at 37 °C with under 5% CO_2_ atmosphere. The next day, the wells with cells were washed with PBS and 1 mL of different concentrations of DOX–PEG–AuNPs–PEG–([^131^I]Tmab) was added, followed by 1.5 h incubation. The medium was then collected, and the cells were gently washed with PBS. Finally, 1 M NaOH was added and the fraction with cells was collected into separate vials. The activity of the collected fractions was measured using a Wizard^2^ Detector gamma counter (Perkin Elmer, Waltham, MA, USA). To calculate the specific binding, the difference between total binding and nonspecific binding was determined (in these wells, HER2 receptors were blocked with a 100-molar excess of unconjugated Tmab).

### 3.10. Internalization Studies

The DOX–PEG–AuNPs–PEG–([^131^I]Tmab) internalization assay was performed on two cell lines, SKOV-3 and MDA-MB-231. Cell preparation for the experiment was identical to the binding experiment (Section 3.9). After removing the medium, 5 nM bioconjugate in 1 mL medium was added and incubated for 1 h at 4 °C. The fraction was collected into vials, the cells were washed twice with PBS, and 1 mL fresh cell medium was added. The plates were incubated for 1 h, 6 h, 18 h, and 24 h. In the next step, the medium was removed and collected, the cells were washed twice with PBS, twice with 0.05 M glycine·HCl pH = 2.8 (to remove bioconjugate bound to the cell membrane but not internalized), and finally with 1 M NaOH to collect the cells (internalized fraction). A 100 M excess of unconjugated Tmab was added to the wells to examine nonspecific binding. All collected fractions were measured using a Wizard^2^ Detector gamma counter.

### 3.11. Confocal Microscopy Imaging

SKOV-3 and MDA-MB-231 cells were seeded in six-well plates (2 × 10^5^ per well) with five sterile 12 mm diameter coverslips (Thermo Fischer Scientific, Waltham, MA, USA) and incubated overnight. Then, after removing the medium, compounds were added and incubated for another 24 h. After this time, 24-well plates were prepared with 1 mL PBS in each well. The coverslips were very gently translated into the wells (1 coverslip per well). A 4% PFA solution was used for fixation and allowed to stand for ca. 10 min. After this time, the solution was removed, and the wells were rinsed twice with PBS. For permeabilization, 0.1% Triton X-100 was added with subsequent washing with PBS after 5 min. Then, 2% BSA + 2% NGS in TBST (blocking buffer, BB) was added and incubated for 30 min, after which the wells were rinsed four times (each rinse was left for 5 min) with TBST solution. Anti-Human IgG was added and incubated for 1 h in the dark, followed by washing three times and the addition of Hoechst 33,258, and the plates were placed in the incubator for 15 min. The wells were rinsed three times with TBST solution, and the coverslips were very carefully and gently moved to the microscope slides (using DAKO Fluorescent Mounting Medium, Agilent, Santa Clara, CA, USA). The slides were allowed to dry and stored in the dark at 4 °C. Wavelengths of 352 nm and 461 nm were used for imaging with Hoechst 33,258 staining, wavelengths of 491 nm and 516 nm were used for Anti-Human IgG staining, and 480 nm and 595 nm wavelengths were used for DOX. Images were also recorded using a bright field through a transmitted light detector (T-PMT). Measurements were performed using FV-1000 confocal microscope (Olympus Corporation, Tokyo, Japan). FV10-ASW 4.02 software (Olympus Corporation) and Fiji (Fiji Is Just ImageJ) [39] were used to analyze the results.

### 3.12. Cytotoxicity Studies

A cytotoxicity assay was performed using the SKOV-3 and MDA-MB-231 cell lines. Cells were seeded in 96-well plates (SKOV-3—2.5 × 10^3^ cells, MDA-MB-231 2 × 10^3^ per well). After 24 h of incubation (at 37 °C, under 5% CO_2_ atmosphere), the medium was extracted and pegylated AuNPs (AuNPs), AuNPs–Tmab, Tmab, AuNPs–DOX, DOX, and the bioconjugate (DOX–AuNPs–Tmab) were added. They were then incubated for 24 h, 48 h, and 72 h, respectively. Before the addition of 20 µL MTS reagent (CellTiter96^®^ Aqueous Non-Radioactive Cell Proliferation Assay), the medium was pulled off, the wells were washed with PBS, and a new medium was added. Absorbance at 490 nm was measured to determine the percentage of cell metabolic activity.

### 3.13. Flow Cytometry—Apoptosis and Cell Cycle Assay

Apoptosis and cell cycle analyses were performed on SKOV-3 cell lines using flow cytometry. Cells were seeded in wells of six-well plates (6 × 10^5^ per well), which were incubated for 24 h at 37 °C under a 5% CO_2_ atmosphere. The wells were then washed and the tested compounds (AuNPs, AuNPs–Tmab, Tmab, AuNPs–DOX, DOX, and DOX–AuNPs–Tmab) were added. Analysis was performed at three time points—after 24 h, 48 h, and 72 h. After the indicated time, cells were treated with Trypsin/EDTA, and the wells were fixed twice with cold phosphate buffer (PBS) and 1X Annexin V Binding Buffer, and 5 µL fluorescin isothiocyanate (FITC) and 5 µL propidium iodide (PI). The tubes with cells were allowed to stand for 15 min in an incubator. 

Cells for cell cycle analysis were prepared using the same conditions, except that the washing of the wells was conducted using cold PBS, and the cells were resuspended in cold 70% ethanol. The samples prepared in this way were placed in a freezer for 1.5 h. Before analysis, cells were centrifuged and washed twice with PBS, and 20 µL propidium iodide (PI) with 2 µL RNase was added.

An FACSCelesta^TM^ instrument (BD Biosciences, San Jose, CA, USA) was used for the above-mentioned assays, and FACSDiva^TM^ v8.0 programming (BD Biosciences, San Jose, CA, USA) was used to analyze the results. The cut-off of measurements was at 10,000 events.

### 3.14. Spheroids

SKOV-3 cells were added to a 96-well ultra-low attachment surface plate (Corning, NY, USA) and cultured for five days, after which AuNPs, AuNPs–Tmab, Tmab, AuNPs–DOX, DOX, and DOX–AuNPs–Tmab were added. Treated spheroids were analyzed over a 30-day period, with medium renewal every 2 days. A Primovert color Axiocam 305 microscope (Zeiss, Jena, Germany) with ZEN 3.0 lite software (Zeiss, Jena, Germany) was used for spheroid analysis.

### 3.15. Statistical Analysis

GraphPad Prism version 8.0 software (GraphPad Software Inc., San Diego, CA, USA) was used for statistical analysis of the experimental data. Student’s *t*-test and one-way ANOVA were used for statistical evaluation, and the results were reported as means with standard deviation. Each experiment was repeated at least three times. The obtained parameters were considered statistically significant when *p* ≤ 0.05, *p* ≤ 0.01, *p* ≤ 0.001, and *p* ≤ 0.0001.

## 4. Conclusions

We have successfully attached DOX and Tmab to AuNPs, which could be potentially utilized for anti-cancer drug targeting. The DOX–PEG–AuNPs–PEG–Tmab bioconjugate was specifically bonded, internalized, and distributed to a peri-nuclear location within HER2-positive cancer cells. Unfortunately, no significant entry of DOX into the cell nucleus was observed, which caused the cytotoxicity of the bioconjugate to be much lower than DOX alone. This effect was observed in in vitro cytotoxicity studies on 2D cultures and was probably related to the blocking of AuNP (30 nm) for DOX entry into the nucleus, which was necessary for the effective action of DOX. Unexpectedly, 3D in-vitro studies revealed that the cytotoxicity of the DOX–PEG–AuNPs–PEG–Tmab bioconjugate did not differ significantly from the cytotoxicity of DOX. Considering that our studies showed similar growth inhibition of 3D cell culture models exposed to DOX and DOX–PEG–AuNPs–PEG–Tmab, targeted therapy seems to be the better option compared to standard chemotherapy. As the cardiotoxic effects of DOX remained a key limiting factor, targeted therapy with the use of this anthracycline and guiding vector Tmab might limit the damage to healthy tissues, well as improve the clinical outcomes of patients suffering from HER2+ cancers. Regarding NP characteristics, including unspecific accumulation in the liver, spleen, or lungs, they can be administrated only locally, i.e., directly into the tumor or resection cavity after surgery. Nevertheless, in vivo experiments in mice are required to verify the effectiveness of the synthesized bioconjugate. It is worth emphasizing that the therapeutic effectiveness of the DOX–PEG–AuNPs–PEG–Tmab conjugate can be significantly increased by external NIR phototherapy or radiofrequency irradiation causing local hyperthermia [40]. Another option is the use of radioactive AuNPs (^198^AuNPs), where three various functions: radiotherapy (^198^Au, β^−^ emitter), chemotherapy (DOX), and guiding vector (Tmab) are combined into one platform. This study has already been started by our research group. It is known from the literature that spherical gold nanoparticles with a size of 30 nm (such as those we studied) have photothermal properties [41]. It is known that in hypoxic cells, the toxic effect of DOX increases with increasing temperature due to better oxygenation of cells.

## Figures and Tables

**Figure 1 molecules-28-02451-f001:**
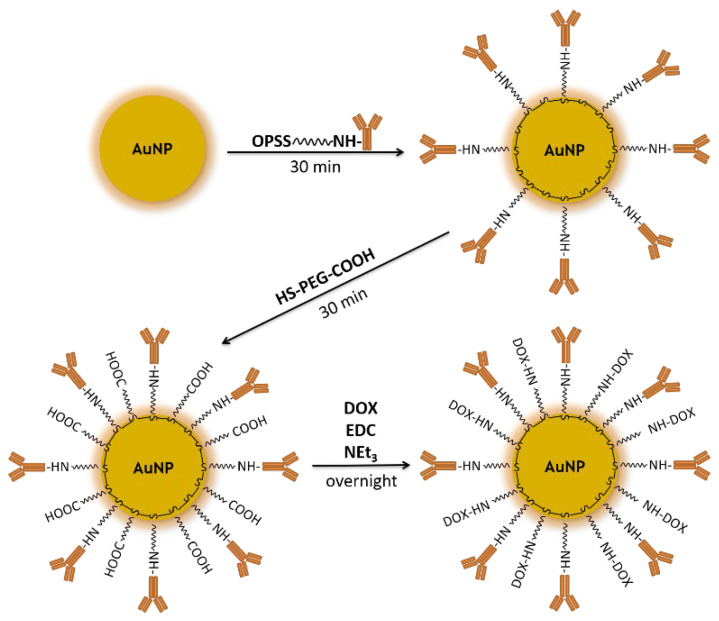
Scheme for the synthesis of the DOX–PEG–AuNPs–PEG–Tmab bioconjugate.

**Figure 2 molecules-28-02451-f002:**
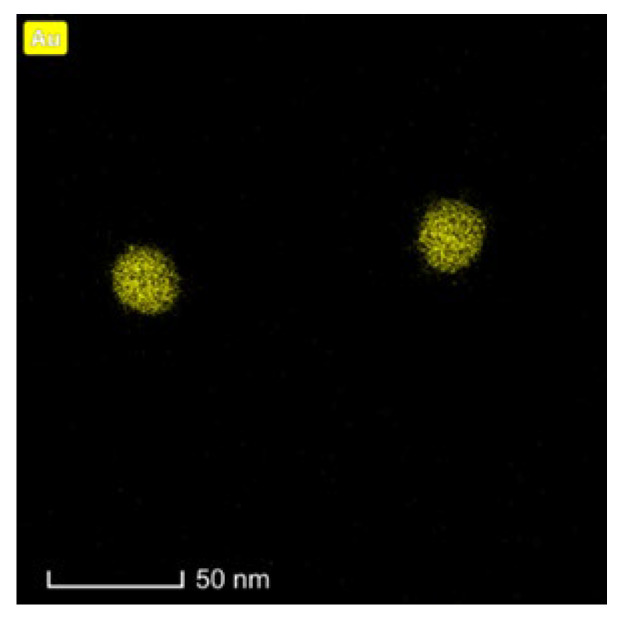
HR-TEM images of AuNPs.

**Figure 3 molecules-28-02451-f003:**
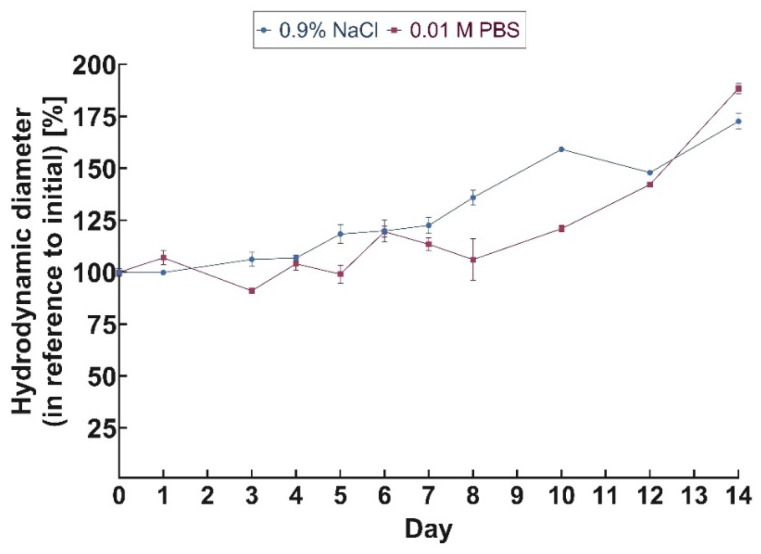
Changes in hydrodynamic diameter of DOX-PEG-AuNPs-PEG-Tmab incubated in two medias—0.9% NaCl and 0.01 M PBS at 37 °C as a function of time.

**Figure 4 molecules-28-02451-f004:**
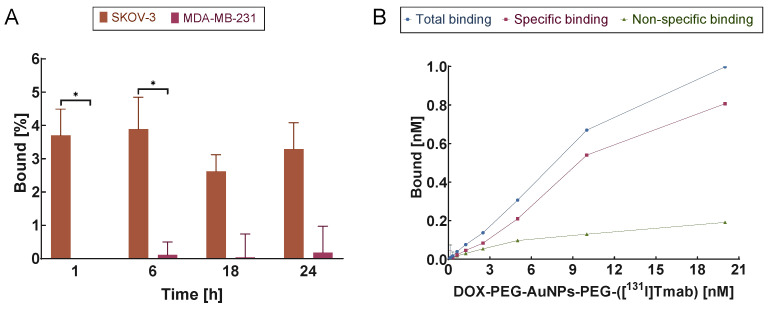
Percentage bioconjugate specificity of DOX–PEG–AuNPs–PEG–([^131^I]trastuzumab) obtained for the SKOV-3 (HER2+) and MDA-MB-231 (HER2−) cell lines (**A**). Data were compared using Student’s *t*-test (* *p* < 0.05). Binding studies of DOX–PEG–AuNPs–PEG–([^131^I]Tmab) on SKOV-3 (HER2+) (**B**).

**Figure 5 molecules-28-02451-f005:**
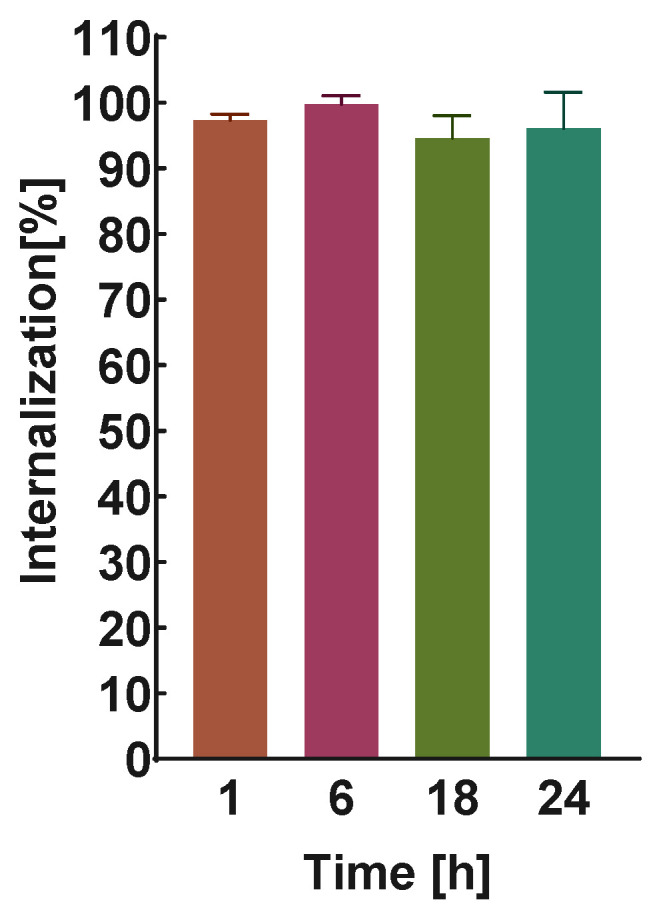
Percentage bioconjugate internalization results obtained for SKOV-3 (HER2+) cells.

**Figure 6 molecules-28-02451-f006:**
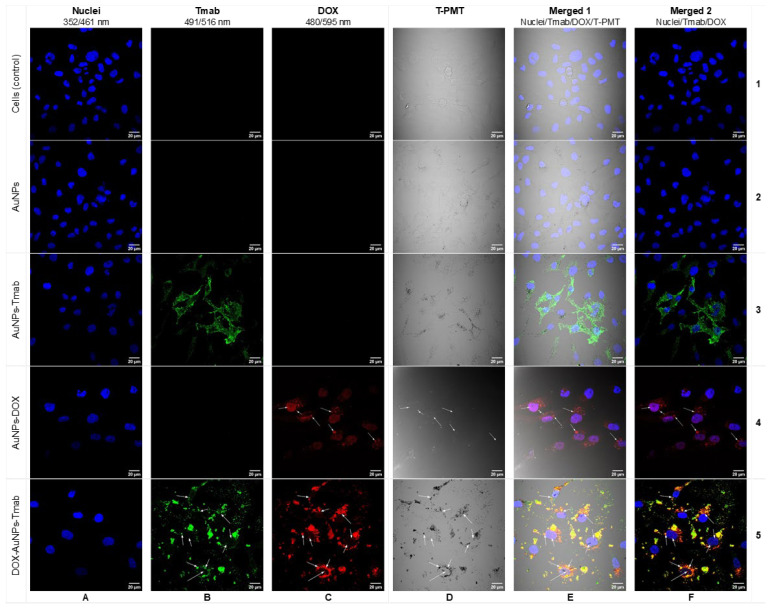
Confocal microscopy images of SKOV-3 cells treated with AuNPs (AuNPs–PEG–COOH), AuNPs–Tmab (AuNPs–PEG–Tmab), AuNPs–DOX (AuNPs–PEG–DOX), DOX–AuNPs–Tmab (DOX–PEG–AuNPs–PEG–Tmab). The first row presents untreated cells (control). Fluorescence signals display: blue—cell nuclei, green—Tmab, red—DOX, black spots—NPs visualized with a transient light detector (T-PMT). Arrows mark the subcellular localization of the AuNPs–DOX and DOX–AuNPs–Tmab.

**Figure 7 molecules-28-02451-f007:**
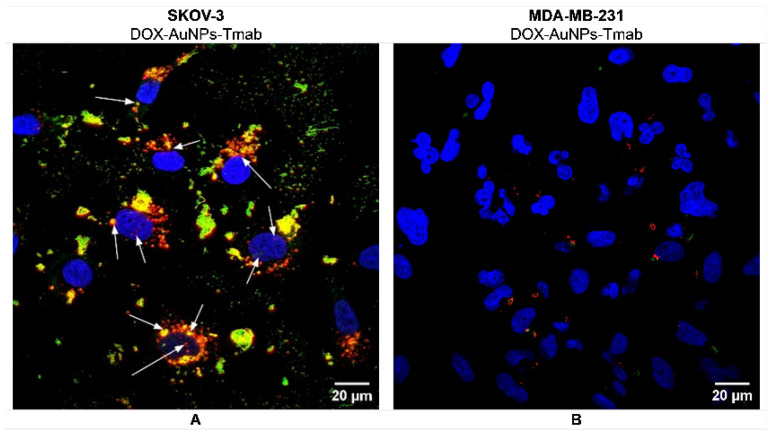
Confocal microscopy images of (**A**) SKOV-3 (HER2+) and (**B**) MDA-MB-231 (HER2−) cells treated with DOX–PEG–AuNPs–PEG–Tmab. Arrows mark the subcellular localization of the DOX–AuNPs–Tmab.

**Figure 8 molecules-28-02451-f008:**
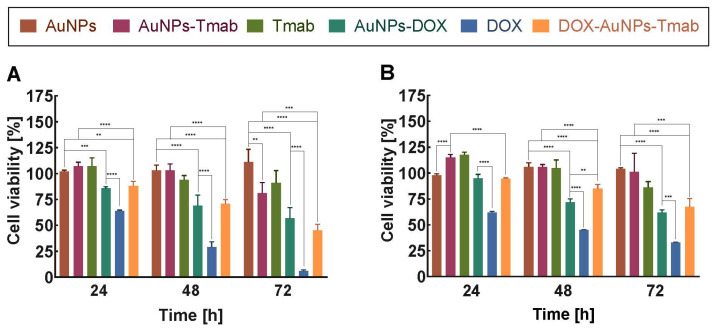
Metabolic activity of SKOV-3 (**A**) and MDA-MB-231 (**B**) cells after treatment with AuNPs, AuNPs–Tmab, Tmab, AuNPs–DOX, DOX, DOX–AuNPs–Tmab. The DOX concentration was 7 µg/mL. Cells were incubated for 24 h, 48 h, and 72 h and then the MTS assay was performed. Untreated cells were used as control (100% viability). Statistics were performed using a one-way ANOVA test comparing AuNPs together with NPs with DOX and/or Tmab attached, and AuNPs–Tmab was compared with Tmab/DOX–AuNPs–Tmab and AuNPs–DOX via DOX/DOX–AuNPs–Tmab. Summarized data represent results from four replicates (*n* = 4; mean ± SD) and were considered significant if *p* ≤ 0.01 (**), *p* ≤ 0.001 (***), or *p* ≤ 0.0001 (****).

**Figure 9 molecules-28-02451-f009:**
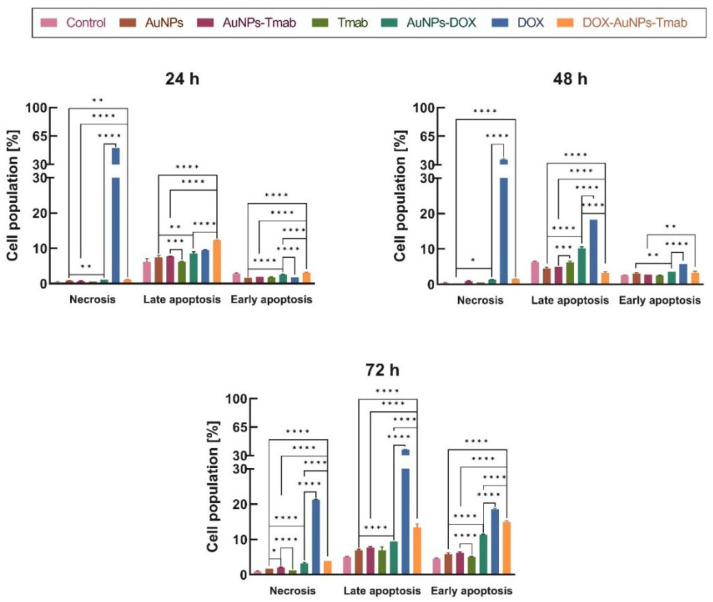
Distribution of SKOV-3 cell populations (necrosis, late and early apoptosis) treated with AuNPs, AuNPs–Tmab, Tmab, AuNPs–DOX, DOX, DOX–AuNPs–Tmab after 24 h, 48 h, and 72 h. The data presented are from four replicates (*n* = 4, mean ± SD). DOX concentration is 7 µg/mL. Untreated cells were used as control, while statistics (one-way ANOVA test) were performed similarly as in the case of the cytotoxicity studies with the use of the MTS assay (Figure 8). Statistical significance was considered significant if *p* ≤ 0.05 (*), *p* ≤ 0.01 (**), *p* ≤ 0.001 (***), or *p* ≤ 0.0001 (****).

**Figure 10 molecules-28-02451-f010:**
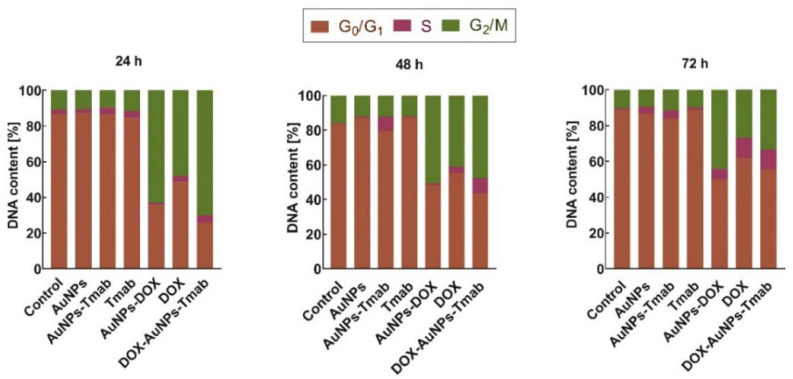
Cycle results of SKOV-3 cells treated with AuNPs, AuNPs–Tmab, Tmab, AuNPs–DOX, DOX, and DOX–AuNPs-Tmab after 24 h, 48 h, and 72 h (*n* = 4 for each data point). DOX concentration is 7 µg/mL. Control cells were not treated with any compounds.

**Figure 11 molecules-28-02451-f011:**
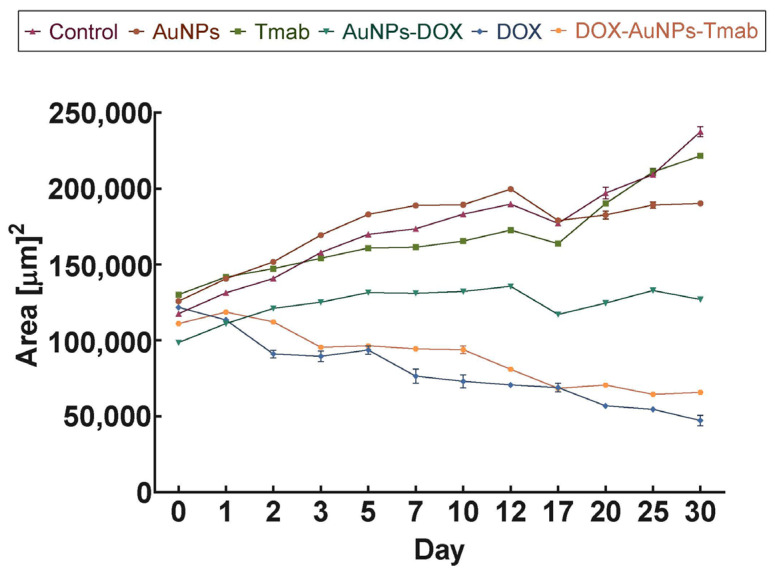
Time-dependent surface development characteristics of SKOV-3 spheroids treated with AuNPs, Tmab, AuNPs–DOX, DOX, and DOX–AuNPs–Tmab or non-treated. Data represent the mean ± SD (*n* = 3).

**Figure 12 molecules-28-02451-f012:**
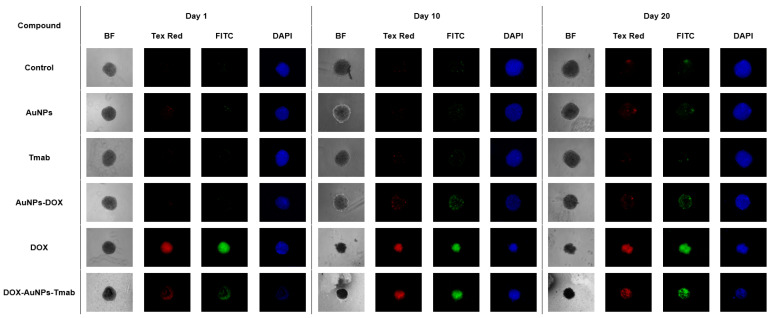
Representative fluorescence microscope images of SKOV-3 spheroids treated with AuNPs, Tmab, AuNPs–DOX, DOX, and DOX–AuNPs–Tmab or non-treated after 1, 10, and 20 days. The first columns (BF) are bright-field images, red signals (Tex Red) are from propidium iodide (PI), green is doxorubicin (FITC), and blue (DAPI) is Hoechst 33,258 staining.

**Table 1 molecules-28-02451-t001:** Results of hydrodynamic diameter, polydispersity index, and zeta potential analysis measurements performed on DLS.

Compound	Hydrodynamic Diameter (nm)	Polydispersity Index (PDI)	Zeta Potential (mV)
AuNPs	35.8 ± 0.5	0.160 ± 0.022	−45.3 ± 1.8
AuNPs–PEG–Tmab	60.2 ± 1.6	0.259 ± 0.001	−38.0 ± 4.0
DOX–PEG–AuNPs–PEG–Tmab	79.9 ± 4.4	0.259 ± 0.017	−38.3 ± 1.2

## Data Availability

Not applicable.

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
