# Peer review of "Doxorubicin- and Trastuzumab-Modified Gold Nanoparticles as Potential Multimodal Agents for Targeted Therapy of HER2+ Cancers"

_molecules, 2023, doi:10.3390/molecules28062451_

Round 1

Reviewer 1 Report

The manuscript of Żelechowska-Matysiak and colleagues describes the development of a gold colloidal carrier with doxorubicin and trastuzumab for HER2 positive breast cancer chemotherapy. The manuscript is well-written and is easy to read. However, the authors adopted several strategies that ended up limiting the potential of the work. It is not scientifically solid or complete, and the conclusion section emphasizes this.

1.      The function of the gold nanoparticles is not well explained. Despite their (already known) advantages, they act solely as carriers and do not bring anything novel to the work.

2.      The authors focused the application on HER2 positive breast cancer. Despite being HER2 positive, SKOV-3 cells are of ovarian origin and present substantial differences from breast cancer cells.

3.      Similarly, the function of doxorubicin is not well explained. When being part of the system, it does not contribute with significant cytotoxicity. Why use it? Despite the known adverse effects, circumvented by Doxil, doxorubicin alone is more effective than the system itself.

4.      The synthesis of the different building blocks is not well characterized. The authors should elucidate this with RMN or other relevant techniques. DLS alone is not sufficient. What are the yields of the reactions?

5.      It is not clear how gold and doxorubicin were quantified. “knowing the mass of gold used” is not an accurate methodology. What is the drug loading and binding efficiency of doxorubicin?

6.      It is not clear if the system is monodisperse, as polydispersity indices are not presented (Table 1).

7.      The stability of the carrier in ‘physiological media’ was performed without the impact of proteins, to which the authors justify as being incompatible with DLS. This assay reflects the stability of the carrier prior to an administration, but not the impact of physiological media. Why did not the authors evaluate this by HR-TEM?

8.      The highest binding affinity for the carrier was 3.7%, which is in agreement with the results obtained by the group (references 19-21). How do the authors justify such a low binding?

9.      Despite the low binding, nanoparticle uptake is close to 100%. How do the authors explain this?

10.   Tumor spheroids are models that better represent cancer growth and development. It would have been more interesting to see them employed instead of the 2D models (cytotoxicity, cell uptake, cell cycle analysis, etc).

Minor:

1.      Full description of RES (reticuloendothelial system is missing).

2.      “Isolated Tmab from herceptin”. Was the medicine used as is, or submitted to processing for antibody isolation?

3.      Section 3.3 Radionuclide should be presented with more detail.

4.      Quantity of NEt3 should be indicated in nmol.  

5.      “The reaction was carried out overnight, and then the unconjugated products were centrifuged, and resuspended in 1 mL of ultrapure deionized water.” This sentence is not correct.

6.      Methods should clearly indicate the number of independent replicates in each experiment.

7.      Number of events in flow cytometry should be indicated.

8.      “3D in-vitro studies (spheroids mimicking the natural tumor microenvironment)”. This information is exaggerated.

9.      “As the cardiotoxic effects of DOX remained a key limiting factor, targeted therapy with the use of this anthracycline and guiding vector Tmab limited damage of healthy tissues”. This information is exaggerated, as there is no data regarding this. 

Author Response

We would like to thank Reviewer very much for all the corrections and comments. We corrected the manuscript according to suggestions and we believe that after that improvements the paper is much more clear and better to read.

Review 1

Comments and Suggestions for Authors

The manuscript of Żelechowska-Matysiak and colleagues describes the development of a gold colloidal carrier with doxorubicin and trastuzumab for HER2 positive breast cancer chemotherapy. The manuscript is well-written and is easy to read. However, the authors adopted several strategies that ended up limiting the potential of the work. It is not scientifically solid or complete, and the conclusion section emphasizes this.

  1. The function of the gold nanoparticles is not well explained. Despite their (already known) advantages, they act solely as carriers and do not bring anything novel to the work.

You are right. Gold nanoparticles (AuNPs) in this article are used as carriers for chemotherapeutic-doxorubicin and guiding vector – Trastuzumab. This is the first publication were these two compounds are attached to the surface of AuNPs and used for the application in targeted drug delivery. As already known doxorubicin is very toxic, causes extremely serious adverse effects, therefore the application of vector can reduce side effects due to targeting only cancer cells and not causing damage of healthy tissues. So, the use of AuNPs as carriers is very important and significant in innovative cancer therapy, especially they exhibit good biocompatibility and controllable biodistribution patterns.

At present, we use Au nanoparticles only as a carrier, but in the future we can apply the properties of Au nanoparticles to induce hyperthermia (NIR laser or radiofrequency) or use radioactive 198Au nanoparticles.

  1. The authors focused the application on HER2 positive breast cancer. Despite being HER2 positive, SKOV-3 cells are of ovarian origin and present substantial differences from breast cancer cells.

SKOV-3 cells were selected for these studies because of its relatively high expression of HER2 receptors, easy culturing and our more experience with these cell line. Of, course we agree, that they are ovarian and present substantial differences from breast cancer cells.

Based on literature data SKOV-3 cells exhibit lower level of HER2 receptors (1.63 × 106)* than BT474 breast cancer cells (HER2+) (2.75 × 106)**, therefore we can predict that our results obtained for SKOV-3 cells would be even better for breast cancer cells - BT474.

Nevertheless we made some corrections in the manuscript including the title.

* Jiang D, Im HJ, Sun H, Valdovinos HF, England CG, Ehlerding EB, Nickles RJ, Lee DS, Cho SY, Huang P, Cai W. Radiolabeled pertuzumab for imaging of human epidermal growth factor receptor 2 expression in ovarian cancer. Eur J Nucl Med Mol Imaging. 2017 Aug;44(8):1296-1305. doi: 10.1007/s00259-017-3663-y. Epub 2017 Mar 6. PMID: 28265738; PMCID: PMC5471126.

** Hathaway HJ, Butler KS, Adolphi NL, Lovato DM, Belfon R, Fegan D, Monson TC, Trujillo JE, Tessier TE, Bryant HC, Huber DL, Larson RS, Flynn ER. Detection of breast cancer cells using targeted magnetic nanoparticles and ultra-sensitive magnetic field sensors. Breast Cancer Res. 2011 Nov 3;13(5):R108. doi: 10.1186/bcr3050. PMID: 22035507; PMCID: PMC3262221.

  1. Similarly, the function of doxorubicin is not well explained. When being part of the system, it does not contribute with significant cytotoxicity. Why use it? Despite the known adverse effects, circumvented by Doxil, doxorubicin alone is more effective than the system itself.

Doxorubicin as was mentioned is very toxic chemotherapeutic, therefore the application of vector can reduce side effects due to targeting only cancer cells and not causing damage of healthy tissues. Using a vector we can also decrease the doxorubicin dose obtaining the same cytotoxic effect.

It is worth to mention that our synthesized drug can only be administered locally - directly to the tumor or to the cavity after cancer resection.

In our case, probably the cytotoxic effect of free DOX in comparison to AuNPs-DOX-Tmab is higher due to the strong amid bond between DOX and AuNPs blocking DOX release. Nevertheless, experiments performed on 3D cell cultures show almost similar cytotoxicity of DOX and  AuNPs-DOX-Tmab.

  1. The synthesis of the different building blocks is not well characterized. The authors should elucidate this with RMN or other relevant techniques. DLS alone is not sufficient. What are the yields of the reactions?

The binding of trastuzumab to AuNPs was confirmed by radiometric method. The method using 131I-trastuzumab allows us to precisely determine the number of trastuzumab molecules attached to Au nanoparticles. We have the possibility to work with radionuclides, therefore we chose this technique. Doxorubicin attachment was determined by UV-Vis, which is very common method in this type of studies. The yield of reaction was very high, η = 95.0± 3.5%.

  1. It is not clear how gold and doxorubicin were quantified. “knowing the mass of gold used” is not an accurate methodology. What is the drug loading and binding efficiency of doxorubicin?

Doxorubicin attachment efficiency was measured using a common UV-Vis spectrophotometric method. Before measurements, calibration curve for doxorubicin was performed. Subsequently, the supernatants, which contained unconjugated doxorubicin, were examined. From this, the yield and amount of attached chemotherapeutic agent were calculated. The yield of reaction was 95.0 ± 3.5%. We added this information to the manuscript.

  1. It is not clear if the system is monodisperse, as polydispersity indices are not presented (Table 1).

We agree. We added PDI to Table 1.

  1. The stability of the carrier in ‘physiological media’ was performed without the impact of proteins, to which the authors justify as being incompatible with DLS. This assay reflects the stability of the carrier prior to an administration, but not the impact of physiological media. Why did not the authors evaluate this by HR-TEM?

Our stability studies with the use of AuNPs were always determined by DLS through the measurement of hydrodynamic diameter over time (e.g. K. Wawrowicz et al. “Au@Pt Core-Shell Nanoparticle Bioconjugates for the Therapy of HER2+ Breast Cancer and Hepatocellular Carcinoma. Model Studies on the Applicability of 193mPt and 195mPt Radionuclides in Auger Electron Therapy”). Moreover, the use of HR-TEM for such experiments is associated with certain limitations, such as the lack of direct access to the HR-TEM device at our Institute.

Thank you for the comment. Maybe we will do it in the future.

  1. The highest binding affinity for the carrier was 3.7%, which is in agreement with the results obtained by the group (references 19-21). How do the authors justify such a low binding?

Such a low binding is typical binding in the case of various proteins.

It is very difficult to find in the literature graphs showing the percentage of binding, mostly in the articles we can only find  such graphs as presented in our publication (Figure 4B). However based on our strong experience in the synthesis of various compounds for targeted therapies of cancers and also literature data this binding is representative and in agreement with other studies.

  1. Despite the low binding, nanoparticle uptake is close to 100%. How do the authors explain this?

It is not uptake but internalization. In this experiment, internalization is estimated as a percentage of bounded bioconjugate which was internalized into the cytoplasm. It means that more than 98% (98% from 3.7%- this is 100% of total binding) of bioconjugate is inside the cell after 1h. The rest 2% is on membrane outside the cell.

  1. Tumor spheroids are models that better represent cancer growth and development. It would have been more interesting to see them employed instead of the 2D models (cytotoxicity, cell uptake, cell cycle analysis, etc).

Cytotoxicity studies were performed on 3D models but others like cell cycle or apoptosis would be very difficult to analyzed by flow cytometry or even binding/internalization using radiometric method. For 3D cell models studies, completely different experiments (e.g. receptor autoradiography) should be performed and specialized devices are required. We regret we were not able to perform such experiments in vitro. Nevertheless, in vivo studies are the best option and they are ongoing with the use of radioactive 198AuNPs.

 Minor:

  1. Full description of RES (reticuloendothelial system is missing).

We added. Thank you.

  1. “Isolated Tmab from herceptin”. Was the medicine used as is, or submitted to processing for antibody isolation?

Tmab was isolated from Herceptin by centrifugation with the use of cut-off membrane (100 kDa). 

  1. Section 3.3 Radionuclide should be presented with more detail.

We added few sentences.

  1. Quantity of NEt3 should be indicated in nmol.  

We added, thank you.

  1. “The reaction was carried out overnight, and then the unconjugated products were centrifuged, and resuspended in 1 mL of ultrapure deionized water.” This sentence is not correct.

We corrected the sentence. “The reaction was carried out overnight, then the unconjugated products were centrifuged and removed from the vial. Further, the conjugated product was resuspended in 1 mL of ultrapure deionized water”.

  1. Methods should clearly indicate the number of independent replicates in each experiment.

We added the sentence to the statistical analysis paragraph: “Each experiment was at least three times repeated”.

  1. Number of events in flow cytometry should be indicated.

In agreement with the gold standard of flow cytometry, the cut-off was at 10 000 events. We added this information.

  1. “3D in-vitro studies (spheroids mimicking the natural tumor microenvironment)”. This information is exaggerated.

Maybe you are right, there are no any blood vessels etc., but in many publications this information is commonly used. We based on literature data. We deleted this sentence “spheroids mimicking the natural tumor microenvironment”.

  1. “As the cardiotoxic effects of DOX remained a key limiting factor, targeted therapy with the use of this anthracycline and guiding vector Tmab limited damage of healthy tissues”. This information is exaggerated, as there is no data regarding this. 

W changed the sentence for “As the cardiotoxic effects of DOX remained a key limiting factor, targeted therapy with the use of this anthracycline and guiding vector Tmab might limit damage of healthy tissues”.

Reviewer 2 Report

Żelechowska-Matysiak et al. described a novel bioconjugate (DOX-AuNPs-Tmab)

consisting of gold nanoparticles attached to the chemotherapeutic agent

doxorubicin and monoclonal antibody – trastuzumab, which exhibited specific

binding to HER2 receptors. Nps showed high specificity of binding to the HER2

receptors and internalization capabilities. Cytotoxicity experiments revealed

a decrease in metabolic activity of cancer cells and surface area reduction of

spheroids treated with DOX-AuNPs-Tmab. The obtained results suggest that

DOX-AuNPs-Tmab has great potential for targeted therapy of HER2

positive tumors. Despite the convincing evidence regarding the effectiveness

of the proposed particles, a number of questions and comments arose for the authors.

1. The authors write that "After only 1 h incubation of SKOV-3 cell line with

radiocompound, the percentage of binding was the highest (~3.7%), whereas

after 6 h decrease in binding was observed." (lines 157-159), although in fig. 4,

one can see that the histogram bar for "6 h" is higher than for "1 h".

2. Starting from Fig. 7, there is a discrepancy between the numbers of the

figures indicated in the text and in the captions to the figures.

3. In Fig. 6 and 7 is not indicated what the arrows represent.

4. How do the authors explain the presence of the Dox signal in the MDA-MB-231

cells (line 207)? Was the signal co-localized with the signal from Tmab, which

would be logical since MDA-MB-231 cells express HER2 at the level of normal

epithelial tissues? Or was it a signal from doxorubicin, which for some reason is

split off from the particles?

5. How can one explain the large accumulation of signals in the upper right corner,

where there are no nuclei (and, accordingly, cells), in line 5 in Fig. 6 and in Fig. 7A?

If this is the sorption of particles on glass, why was it not observed for MDA-

MB-231 (Fig. 7B)?

6. The "materials and methods" describe a method for estimating the number

of trastuzumab molecules bound to particles, but do not provide a method for

estimating the loading of NPs with doxorubicin, although effective concentrations

for doxorubicin are given.

7. How can the authors explain the noticeable decrease in the size of spheroids

of each type, including control ones, on the 17th day of the experiment (Fig. 11

in the text, Fig. 10 in the captions, line 313)?

8. It would seem that the low-molecular substance doxorubicin penetrates

spheroids more easily than particles. How can the authors substantiate the fact that,

over time, the sizes of spheroids treated with pure doxorubicin and their particles

behaved in the same way?

9. In the discussion, it would be great to see a more complete comparison of

the proposed particles with analogues described in the literature (in terms of

doxorubicin loading, biocompatibility, maximum cytotoxic effect).

10. The authors suggest a possible combination therapy with the proposed

particles, using the photothermal properties of the gold particles. It is known that

absorption in the NIR region strongly depends on the shape of gold particles. Did

the authors study the absorption spectra of their particles and their

ability to photothermia?

In general, this work can be a good start for future in vivo studies, and the

manuscript itself, after finalizing these comments, can be published in the journal

"Molecules".

Author Response

We would like to thank Reviewer very much for all the corrections and comments. We corrected the manuscript according to suggestions and we believe that after that improvements the paper is much more clear and better to read.

Review 2

Å»elechowska-Matysiak et al. described a novel bioconjugate (DOX-AuNPs-Tmab) consisting of gold nanoparticles attached to the chemotherapeutic agent doxorubicin and monoclonal antibody – trastuzumab, which exhibited specific  binding to HER2 receptors. Nps showed high specificity of binding to the HER2  receptors and internalization capabilities. Cytotoxicity experiments revealed  a decrease in metabolic activity of cancer cells and surface area reduction of  spheroids treated with DOX-AuNPs-Tmab. The obtained results suggest that  DOX-AuNPs-Tmab has great potential for targeted therapy of HER2  positive tumors. Despite the convincing evidence regarding the effectiveness  of the proposed particles, a number of questions and comments arose for the authors.

  1. The authors write that "After only 1 h incubation of SKOV-3 cell line with radiocompound, the percentage of binding was the highest (~3.7%), whereas after 6 h decrease in binding was observed." (lines 157-159), although in fig. 4, one can see that the histogram bar for "6 h" is higher than for "1 h".

Thank you for this comment, we have changed this information in the article. This is obviously our mistake, it should be 'after 18 h' instead of ‘6 h’.

  1. Starting from Fig. 7, there is a discrepancy between the numbers of the figures indicated in the text and in the captions to the figures.

We corrected, thank you.

  1. In Fig. 6 and 7 is not indicated what the arrows represent.

Yes, this is true. Thank you for that comment. We added the information: "Arrows mark the subcellular localization of the AuNPs-DOX and DOX-AuNPs-Tmab” (Figure 6). “Arrows mark the subcellular localization of the DOX-AuNPs-Tmab” (Figure 7).

  1. How do the authors explain the presence of the Dox signal in the MDA-MB-231 cells (line 207)? Was the signal co-localized with the signal from Tmab, which would be logical since MDA-MB-231 cells express HER2 at the level of normal epithelial tissues? Or was it a signal from doxorubicin, which for some reason is split off from the particles?

The presence of the DOX signal in MDA-MB-231 cells was co-localized with the Tmab signal. We did not observe the split-off of the chemotherapeutic agent.

  1. How can one explain the large accumulation of signals in the upper right corner, where there are no nuclei (and, accordingly, cells), in line 5 in Fig. 6 and in Fig. 7A? If this is the sorption of particles on glass, why was it not observed for MDA-MB-231 (Fig. 7B)?

In other nanoconjugates like AuNPs-Tmab and AuNPs-DOX we did not observe such signals. Probably, this is due to the fact that the preparation was worse washed out in the well, in comparison to other compounds.

  1. The "materials and methods" describe a method for estimating the number of trastuzumab molecules bound to particles, but do not provide a method for estimating the loading of NPs with doxorubicin, although effective concentrations for doxorubicin are given.

Thank you for this comment, we have added this to the text. Doxorubicin attachment efficiency was measured using a common UV-Vis spectrophotometric method (Jasco V-650). Before measurements, a calibration curve for doxorubicin was performed. Subsequently, the supernatants, which contained unconjugated doxorubicin were examined. From this, the yield and amount of attached chemotherapeutic agent were calculated. The binding efficiency of DOX was 95.0 ± 3.5%.

  1. How can the authors explain the noticeable decrease in the size of spheroids of each type, including control ones, on the 17th day of the experiment (Fig. 11 in the text, Fig. 10 in the captions, line 313)?

There is only one our explanation. Probably, the decrease in the size spheroids on the 17th day of experiment is caused by hand-washing spheroids. In order to obtain reliable measurements and good images on the microscope, the spheroids had to be washed as gently as possible and of course we did it lightly. Please note that up to17th day, measurements were performed every 1-3 days. The first longest break happened between days 12 and 17. Probably very gentle washing caused damage of some cells in the spheroids (spheroid was not very compact at this time), making them smaller. However, at subsequent measurement points, no such trend was noticed. Unfortunately, it is really difficult to find very appropriate explanation. We have found in the literature the graph presenting “Tumor volume measured in BT474/multidrug resistance bearing nude mice treated by saline, empty liposome, liposomal bevacizumab, free DOX, liposomal DOX, immunoliposomal DOX, and immunoliposomal DOX + liposomal bevacizumab, where also similar trend was observed (International Journal of Nanomedicine 2017:12, page 677, Figure 6).

  1. How can the authors substantiate the fact that, over time, the sizes of spheroids treated with pure doxorubicin and their particles behaved in the same way?

Based on the Figure 11, the sizes of spheroids treated with pure doxorubicin and their particles behaved similarly. However, if we consider the stained spheroids (Figure 12) we can see that three-dimensional (3D) cells  treated with doxorubicin are a little more damaged than those treated with bioconjugate. It is very difficult to answer clearly.We think in vivo studies in the future are required to confirm our in vitro experiments.

  1. In the discussion, it would be great to see a more complete comparison of the proposed particles with analogues described in the literature (in terms of doxorubicin loading, biocompatibility, maximum cytotoxic effect).

We added more information about similar NPs described in the literature. Unfortunately, it is not easy to find more papers and information regarding the studies performed on the flow cytometer as well as spheroids.

  1. The authors suggest a possible combination therapy with the proposed particles, using the photothermal properties of the gold particles. It is known that absorption in the NIR region strongly depends on the shape of gold particles. Did the authors study the absorption spectra of their particles and their ability to photothermia?

It is known from the literature that spherical gold nanoparticles with a size of 30 nm (such as we studied) have photothermal properties [Zakaria, et al, Lasers Med Sci 31, 625–634 (2016)]. Therefore good idea would be to combine hyperthermia generated on gold nanoparticles with the chemotoxic properties of DOX. It is known that in hypoxic cells, the toxicity of DOX increases with increasing temperature. Maybe in the future we will try to implement this idea. We have added this sentence to the manuscript.

Round 2

Reviewer 1 Report

The authors have followed my recommendations, answered my concerns regarding the use of doxorubicin and gold nanoparticles and critically justified the development of the nanosystem.